# Activin and BMP Signalling in Human Testicular Cancer Cell Lines, and a Role for the Nucleocytoplasmic Transport Protein Importin-5 in Their Crosstalk

**DOI:** 10.3390/cells12071000

**Published:** 2023-03-24

**Authors:** Karthika Radhakrishnan, Michael Luu, Josie Iaria, Jessie M. Sutherland, Eileen A. McLaughlin, Hong-Jian Zhu, Kate L. Loveland

**Affiliations:** 1Centre for Reproductive Health, Hudson Institute of Medical Research, 27-31 Kanooka Grove, Clayton, VIC 3168, Australia; 2Department of Surgery, Royal Melbourne Hospital, University of Melbourne, Parkville, VIC 3050, Australia; 3Priority Research Centre for Reproductive Science, Schools of Biomedical Science & Pharmacy and Environmental & Life Sciences, University of Newcastle, Callaghan, NSW 2305, Australia; 4Hunter Medical Research Institute (HMRI), New Lambton Heights, NSW 2305, Australia; 5Faculty of Science, Medicine and Health, University of Wollongong, Gwynneville, NSW 2500, Australia; 6Department of Molecular and Translational Sciences, Monash University, Clayton, VIC 3800, Australia

**Keywords:** testicular germ cell tumour, Importin, TGF-β superfamily, seminoma, non-seminoma

## Abstract

Testicular germ cell tumours (TGCTs) are the most common malignancy in young men. Originating from foetal testicular germ cells that fail to differentiate correctly, TGCTs appear after puberty as germ cell neoplasia in situ cells that transform through unknown mechanisms into distinct seminoma and non-seminoma tumour types. A balance between activin and BMP signalling may influence TGCT emergence and progression, and we investigated this using human cell line models of seminoma (TCam-2) and non-seminoma (NT2/D1). Activin A- and BMP4-regulated transcripts measured at 6 h post-treatment by RNA-sequencing revealed fewer altered transcripts in TCam-2 cells but a greater responsiveness to activin A, while BMP4 altered more transcripts in NT2/D1 cells. Activin significantly elevated transcripts linked to pluripotency, cancer, TGF-β, Notch, p53, and Hippo signalling in both lines, whereas BMP4 altered TGF-β, pluripotency, Hippo and Wnt signalling components. Dose-dependent antagonism of BMP4 signalling by activin A in TCam-2 cells demonstrated signalling crosstalk between these two TGF-β superfamily arms. Levels of the nuclear transport protein, IPO5, implicated in BMP4 and WNT signalling, are highly regulated in the foetal mouse germline. *IPO5* knockdown in TCam-2 cells using siRNA blunted BMP4-induced transcript changes, indicating that IPO5 levels could determine TGF-β signalling pathway outcomes in TGCTs.

## 1. Introduction

The incidence of type II testicular germ cell cancers is increasing worldwide and accounts for roughly 1% of the malignancies affecting young men aged between 15–40 [1,2]. These are understood to arise from foetal germ cells that failed to differentiate into spermatogonia but did not undergo apoptosis as would occur normally in foetal life; instead they are retained in the adult testis as germ cell neoplasia in situ (GCNIS) cells which share gene expression and epigenetic signatures with primordial germ cells and foetal male germline cells (gonocytes) [1,3,4,5,6,7]. Although genetic factors contribute to testicular germ cell tumour (TGCT) risk, inappropriate signals from the somatic environment are understood to be key to their aetiology and progression [8].

TGCTs are generally diagnosed after puberty, when GCNIS cells have transformed into either a seminoma or non-seminoma subtype [1]. Seminoma cells appear homogeneous, resembling primitive germ cells and GCNIS. In contrast, the stem components of non-seminomas, known as embryonal carcinomas (EC), are pluripotent; they are sometimes described as malignant embryonic stem cells (ESCs) because they can differentiate into all three germ layers and extra embryonic tissue, presenting as teratomas, yolk sac tumours, and choriocarcinomas [1]. It has been proposed that the formation of seminomas from GCNIS is a default pathway, so that the formation of non-seminomas requires the reprograming or reactivation of pluripotency factors [9,10]. Seminomas can eventually differentiate into non-seminomas with or without the formation of an EC intermediate cell type [11]. Based on the finding that the *KIT* gene is mutated in some seminomas but appears to be wild type in non-seminomas, it has been proposed that only seminoma cells with wild type *KIT* can transform into non-seminoma [12]. These findings collectively suggest that the emergence of a particular TGCT subtype may reflect both how and at what stage germ cell development is disrupted.

The conditions contributing to foetal germ cell transformation into GCNIS and their further progression into seminomas or non-seminomas are not well understood. Several TGF-β (transforming growth factor-β) superfamily ligands influence the function of developing somatic cells in mice during the equivalent developmental window of relevance to GCNIS emergence in the foetal human testis. In particular, activin A levels are known to determine steroid production, affect testis cord and vasculature formation, and influence foetal germ cell differentiation [13,14,15,16]. Due to the varied roles of TGF-β superfamily cytokines during testicular germline and somatic cell differentiation [17,18], their potential roles in testicular germ cell tumour aetiology have been investigated [18,19,20,21,22,23]. In particular, mutations in an ortholog of the human *BMPR1B* gene resulted in the formation of seminoma-like tumour in zebrafish [23,24]. In humans, single nucleotide polymorphisms in *TGFB1* gene are associated with an increased risk of developing testicular cancer [25]. Gene expression profiling studies identified a reduction in BMP signalling activity in undifferentiated tumours, such as seminomas and dysgerminomas, as compared to more differentiated tumours, such as yolk sac tumours [22]. Overall, these studies have implicated both arms of the TGF-β signalling pathway in the pathogenesis of testicular cancer and indicate that BMP signalling activities and factors that govern them are of clinical importance.

The TCam-2 seminoma-like cell line has been an important model for investigations of TGCT biology, and it shares many features with gonocytes, as would be expected [26]. TCam-2 cells can be reprogrammed by a somatic microenvironment into embryonal carcinoma-like cells following long-term culture and transplantation [27,28]. Importantly, inhibition of BMP signalling and of the BMP target gene, *SOX17*, enable the upregulation of Nodal signalling [28,29]. This may be fundamental to the early events that reprogram TCam-2 cells into an EC-like cell and by extension reflect processes in the foetal testis that lead to TGCT emergence. Similarly, activin A exposure promotes a less differentiated phenotype in short-term cultures of TCam-2 cells [30]. Thus, understanding the complex network of concurrent TGF-β superfamily signals and its regulation may reveal how cells with gonocyte characteristics survive into adulthood in young men and transform into TGCTs.

The initiation of TGF-β signalling by the binding of a TGF-β ligand to cell membrane receptors results in the recruitment and phosphorylation of rSMADs (receptor activated SMADs (Suppressor of Mothers Against Decapentaplegic)), typically either SMAD2 and/or SMAD3 in response to TGF-βs/activins/Nodal or SMAD1, SMAD 5, and/or SMAD9 in response to the BMPs (reviewed in [31]). The trimeric complex formed when two phosphorylated SMADs combine with a SMAD4 molecule is transported by importin proteins into the nucleus where SMADs effect changes in gene transcription by acting in concert with various nuclear co-factors [32]. A key feature of this pathway is the shared engagement of a relatively small number of receptors and intracellular signal transduction molecules by a large number of different ligands. The importance of ligand competition for receptor access has been recently reviewed, with the relevance of this to diverse normal and pathological conditions addressed [33]. Here, it is also important to note that in addition to transducing signals from TGF-β signalling pathway, SMADs also participate in extensive crosstalk with other signalling pathways, such as Notch, Wnt, and fibroblast growth factor receptors (FGFR). In addition, mitogen-activated protein (MAP), phosphoinositide 3-kinase (PI3K)-Akt, nuclear factor κB (NF-κB), and Janus kinase/signal transducers and activators of transcription (JAK/STAT) pathways [34,35,36], many of which also play a crucial role during germ cell development [37] are also implicated.

Importins are of central importance to many cellular functions because they selectively bind cargo; best studied for their roles in facilitating protein transport into the nucleus through nuclear pores, importins also anchor binding partners in particular subcellular domains to regulate developmental switches and cellular phenotypes [38]. Amongst the seven different IMPα proteins in humans and 20 IMPβs in mammals, the tightly regulated expression of several has been documented during development, including in embryonic stem cells and muscle cells, as well as in the germ and somatic cells of the testis [39,40,41]. Importin 5, *IPO5*, (previously named *RANBP5* and *KPNB3*), is a highly conserved member of IMPβ family that is strikingly abundant in foetal mouse germ cells and has a sex-specific subcellular localization pattern after sex determination [42]. Known to transport cargo linked to Wnt and Notch signalling, IPO5 was also shown to selectively transport the BMP signalling mediators, SMADs 1/5/9, but not the TGFβ/activin/Nodal SMADs, SMAD2/3, into the nucleus [43]. This is particularly important in the context of recent studies which have revealed the importance of the balance between TGF-β-Smad2/3 and BMP-Smad1/5 activities in determining the outcome of tumour metastasis in several models [44]. We previously reported *IPO5* expression in GCNIS and seminomas and in the seminoma-derived TCam-2 cell line [45]. The ablation of *IPO5* using siRNA reduced BMP4-induced *SNAI2* expression in TCam-2 cells [45], revealing a link between IPO5 levels and BMP4 signalling outcomes that may regulate the levels of an important transcription factor of known general importance to tumour metastasis. However, the role of *IPO5* in TGCT progression and metastasis is otherwise unknown and may relate to other processes, such as proteasome formation and localization [46]. An inspection of the GeneCards database revealed *IPO5* to be highly expressed in the NT2/D1 embryonic carcinoma cell line; this provides an additional point of interest, as *IPO5* is transiently upregulated in differentiating embryonic stem cells [47] and may therefore be involved in the transition between developmental fates. On this basis, we hypothesized that IPO5 levels determine the outcomes of TGF-β superfamily signals that are of key importance to developmental switches in male germ cells in normal and pathological states.

The accumulating evidence for the importance of TGF-β superfamily signalling activities in testicular cancer provides impetus for understanding potential cross-talk between the two arms of this pathway that could determine TGCT pathologies. With the ultimate goal of understanding how TGF-β superfamily ligands influence testicular cancer progression, this study examines activin A and BMP4 signalling in the seminoma-derived TCam-2 cell line and in the NT2/D1 cell line representing non-seminoma TGCTs. RNA sequencing, bioinformatic analyses and qRT-PCR were used to identify and validate target genes and to assess the major signalling pathways they affected. The potential for other factors to impact the activities of activin A and BMP4 is addressed, with points of overlap and distinction indicated. A dose-dependent suppression of BMP signalling activation by activin A was documented in both TCam-2 and NT2/D1 cells, and the impact of *IPO5* knockdown on BMP4 signalling in TCam-2 cells was demonstrated for selected target genes. These findings advance our knowledge of potential mechanisms by which input from both arms of the TGF-β signalling pathway may influence the establishment and outcomes of TGCTs.

## 2. Materials and Methods:

### 2.1. Cell Lines and Growth Factor Treatments

The TCam-2 human testicular seminoma cell line (originally provided by Prof. Sohei Kitazawa) [26] was maintained in RPMI 1640 medium (Gibco, Thermo Fisher Scientific, Waltham, MA, USA) supplemented with 10% foetal bovine serum (FBS, Bovogen, Keilor East, Australia) and 0.5% penicillin/streptomycin (Gibco, Thermo Fisher Scientific, Waltham, MA, USA) at 37 °C in 5% CO_2_. The NT2/D1 human embryonal carcinoma cell line (provided by Prof. Andrew Sinclair) [48] was maintained in DMEM GlutaMAX^TM^ medium (Gibco, Thermo Fisher Scientific, Waltham, MA, USA) supplemented with 10% FBS and 0.5% penicillin/streptomycin at 37 °C in 5% CO_2_. The culture medium was replaced every two days and the cells were passaged when 80–85% confluent.

For growth factor exposure experiments, TCam-2 and NT2/D1 cells were serum starved overnight (12–14 h), followed by stimulation with either activin A (R & D Systems, Minneapolis, MN, USA) or BMP4 at 5 ng/mL (low dose) or at 25 ng/mL (high dose) in serum-free medium for 6 or 24 h. To investigate potential synergy or antagonism, cells were stimulated with a combination of activin A and BMP4 at low and high doses. An equivalent volume of diluent (0.4% BSA in 4 mM HCl) was used as a vehicle control in every experiment. Experimental replicates are indicated in figure legends.

### 2.2. RNA Sequencing and Bioinformatic Analysis

Following cell line exposure to growth factors, RNA was extracted using TRIzol (Sigma-Aldrich, Merck, St. Louis, MO, USA) and treated with DNase-free kit (Invitrogen, Thermo Fisher Scientific, Waltham, MA, USA) as per the manufacturer’s specifications. Duplicate samples were prepared for each cell line and treatment, and this was repeated once in order to yield a total of 4 samples for each condition. RNA sequencing was performed on the DNBSEQ-G400 with PE100 at *Beijing Genomics Institute* (BGI, Shenzhen, China). Initial bioinformatic analyses was performed by staff at the Monash University Bioinformatics Platform. Adaptor sequence and low read contamination were removed and filtered results were provided as a FASTAQ file format. Files were analysed through the RNAsik (v1.5.4) pipeline [49] using STAR aligner [50] in conjunction with the human reference genome (GRCh38). Raw counts were analysed using Degust [51], a web interface where differential expression data were analysed with respect to limma–voom normalisation. Differentially expressed transcriptomic data sets across both technical and independent experiments were identified using a false-discovery rate (FDR) of <0.05 and a >3-fold change (log_2_FC ≥ ±1.5) in expression level. The genes identified as differentially expressed were functionally annotated using the Database for Annotation, Visualization, and Integrated Discovery (DAVID) online tool. Ingenuity pathway analysis (IPA) software (Qiagen, Thermo Fisher Scientific, Waltham, MA, USA) was used to identify potential upstream regulators. Z-scoring was used to determine the predicted activation score or inhibition score; Z score cut off of ±1.8 was employed.

### 2.3. cDNA Synthesis and qRT-PCR

First-strand cDNA synthesis was performed with 500 ng of DNAse-treated RNA, 50 μM random hexamers (Promega, Madison, WI, USA), 200 units/μL of Superscript III Reverse Transcriptase (Invitrogen, Thermo Fisher Scientific, Waltham, MA, USA), and 10 μM of dNTPs (Sigma-Aldrich, Merck, St. Louis, MO, USA). RNA denaturation was carried out at 65 °C for 5 min, primer extension at 50 °C for 1 h, and enzyme inactivation at 70 °C for 15 min. Negative control reactions lacking Superscript III were included for each sample. 

Quantitative real-time PCR (qRT-PCR) reactions were performed using diluted cDNA template (1:20), Power SYBR-Green PCR master mix (Thermo Fisher Scientific, Waltham, MA, USA), and specific primer pairs (Table 1; Integrated DNA Technologies, Coralville, IA, USA). The qPCR primers were designed using Primer-BLAST software (https://www.ncbi.nlm.nih.gov/tools/primer-blast/, accessed on 31 September 2022). qPCR was performed on the Applied Biosystems 7900HT Sequencing Detection machine (Applied Biosystems, Thermo Fisher Scientific, Waltham, MA, USA) at that Medical Genomics Facility, Monash Health Translation Precinct. The qPCR was carried out at 95 °C for 10 min, with 45 cycles of amplification (95 °C for 15 s) and extension (62 °C for 30 s). Results were analysed using the SDS Automatic Controller 2.3 (Applied Biosystems, Thermo Fisher Scientific, Waltham, MA, USA). Three technical replicates were performed for each sample with each primer pair, and CT values averaged to yield a single data point. The target gene values were normalized to that of the housekeeper, *RPLP0* (ribosomal protein lateral stalk subunit P0). Vehicle control values were set to one, and growth factor-treated values are presented as the fold-change relative to values obtained in the vehicle control.

### 2.4. Dual Promoter Luciferase Assay

The BMP-SMAD1/5 signalling activity reporter, Adenoviral (Ad)-Bre-Firefly (F)-luciferase (Luc) (multiplicity of infection, MOI: 2000), and the activin A-SMAD3 reporter, Ad-CAGA-Gaussia (G)-Luc (MOI: 1000), were produced as previously described [44,52]. To measure the two signalling activities in the same cell, TCam-2 and NT2/D1 cells were simultaneously infected with both adenoviral reporter constructs in a 96-well plate and cultured overnight (12–14 h) with 5000 cells per well. The following morning, cells were exposed to either growth factor(s) or the vehicle in serum-free medium. Luciferase activity was assessed 6 and 24 h after stimulation using the Luciferase Reporter Assay Kit (Promega, Madison, WI, USA). All treatments were carried out in triplicate. Luciferase activity outcomes are presented as relative luciferase units normalized to the basal reporter level. Vehicle control readings were set to 1, and all other readings were normalized to this value. Representative results of at least three independent experiments are shown.

### 2.5. Transient IPO5 Silencing

Cells seeded in 12-well plates at 75,000 cells per well were incubated overnight until 50–60% confluent, then transfected with 12.5 pmol of scramble siRNA (SCRAM) or *IPO5* siRNA constructs (4390771, Thermo Fisher Scientific, Waltham, MA, USA), according to the Life Technologies RNA iMAX lipofection protocol (Thermo Fisher Scientific, Waltham, MA, USA), and incubated for 24 h at 37 °C, 5% CO_2_. To determine the silencing efficiency, RNA was extracted from the transfected cells and qPCR performed using *IPO5* primers (Table 1). A silencing efficiency of 70% or higher was consistently achieved. To examine the impact of *IPO5* silencing on activin A and BMP4 treatment outcomes, transfected cells were stimulated with growth factors for 48 h as described above. All treatments were performed in triplicate. The SCRAM siRNA construct and vehicle were used as transfection and treatment controls, respectively.

### 2.6. Protein Extraction and Immunoblotting

IPO5 levels in TCam-2 and NT2/D1 cell lines were assessed by Western blotting. Cells were washed in cold PBS and lysed in RIPA lysis buffer (10^7^ cells per ml of buffer; 150 mM NaCl, 1% Trition X-100, 0.5% sodium deoxycholate, 0.1% SDS, 50 mM Tris, pH 8) containing Halt™ Protease Inhibitor Cocktail (Thermo Fisher Scientific, Waltham, MA, USA). Proteins were quantitated using the DC Protein Assay Kit (Bio-Rad Laboratories, Hercules, CA, USA). A total of 5 µg of protein was diluted 1:1 in 2X sample buffer (0.125 m Tris-HCl pH 6.8, 4% SDS, 20% glycerol, 0.02% bromophenol blue, 10% β-mercaptoethanol), then denatured for 10 min at 95 °C. Electrophoresis was performed using 10% Mini-PROTEAN^®^ TGX™ Precast Protein Gels (Bio-Rad Laboratories) followed by transfer onto nitrocellulose Hybond C-Extra membrane (GERPN303D, Merck, Thermo Fisher Scientific, Waltham, MA, USA). For protein detection, the membrane was blocked with 5% non-fat instant milk in Tris-buffered saline, 0.1% Tween (TBST), followed by overnight incubation with 4 ug/mL of anti-Importin-5 antibody (GTX32684; GeneTex, Irvine, CA, USA) at 4 °C with rocking. The antibody to α-tubulin at 1:5000 (T5168; Sigma-Aldrich) was used as the loading control, and 800 nm anti-rabbit (926-32211, LI-COR, Lincoln, NE, USA) and 700 nm anti-mouse secondary antibodies (A-21057, Thermo Fisher Scientific, Waltham, MA, USA) were added (1:10000 dilution, 1 h at room temperature) to detect the primary antibodies. All primary and secondary antibodies were diluted in 3% milk in TBST. Images were processed using Li-Cor Odyssey Infrared Imaging System (LI-COR Biosciences, model #9120, Lincoln, NE, USA), and densitometric quantification was performed using ImageJ software [53]. Quantitative metrics extracted from sample bands in each lane were first normalised to the α-tubulin loading control value; the IPO5 signal for each sample is expressed as a fold-change relative to the respective SCRAM control sample.

### 2.7. Statistical Analyses

All graphs were plotted using GraphPad Prism version 9 (San Diego, CA, USA). Values from control versus treated samples from 3 or 4 independent experimental results presented as mean ± SE, unless stated otherwise, are described in each figure legend. A two-tailed *t*-test and ordinary one-way ANOVA followed by Sidak’s multiple comparison test was performed, with *p* < 0.05 determining significance.

## 3. Results

### 3.1. Transcriptome Profiling and Bioinformatic Analyses of Activin A- and BMP4-Treated TCam-2 and NT2/D1 Cells

RNA sequencing was performed to identify and compare the gene expression profiles of TCam-2 and NT2/D1 cells in response to 6 h of activin A or BMP4 treatment. As a starting point, the analysis of untreated TCam-2 and NT2/D1 cells illustrates the presence of signalling molecules (receptors, SMADs) in each (Figure 1).

A gene was considered as differentially expressed if transcript levels were >3-fold different following growth factor exposure, with an FDR value ≤0.05. Functional analyses of differentially expressed genes used the DAVID online tool (in Figure 2, and Appendix A), while potential upstream regulators were identified using Ingenuity Pathway Analysis (Table 2, and Appendix A). Selected differentially expressed transcripts were confirmed using real time PCR on independent samples (Figure 3). 

### 3.2. Activin A Treatment

Activin A exposure altered levels of 347 (1.7%) and 885 (4.4%) transcripts in TCam-2 and NT2/D1 cells, respectively; amongst these, 96 were altered in both cell lines (Figure 2A). The pathway analyses of activin A targets common to both cell lines revealed that activin A treatment significantly upregulated pluripotency, cancer, TGF-β, Notch, p53, and Hippo signalling pathways (Figure 2A). Detailed DAVID pathway analyses are provided as in the, Appendix A. A closer look at individual signalling pathways identified key affected modulators, effectors, and target genes involved (Appendix A). Ras/Rap signalling components, including guanine nucleotide exchange factors (GEFs; *RASGRF1*, *RASGRF2*, *RASGRP1*, *RASA3*, and *RAPGEF*), GTPase-activating proteins (GAPs; SIPAIL2), and effectors involved in endocytosis and cytoskeletal remodelling (*RGL1*, *RIN1*, and *TIAM1*), as well as cell adhesion, polarity, and migration (*TIAM1*, *PRKD1*, and *ARAP3*), were upregulated following activin A stimulation. PI3K-Akt signalling effectors and target genes involved in cell survival (*NR4A1*, *CREB5*, *BCL2L11*, and *MYB*) and cell cycle progression (*CCND1*, *CCNE2*, and *MYC*) were modulated primarily in NT2/D1 cells by activin A treatment. The transcript encoding a key PI3K Akt pathway component, serine/threonine kinase *AKT3*, which was upregulated in activin A-treated TCam-2 cells. Activin A altered P53 signalling pathway components in both lines, including several involved in cell cycle and DNA repair (*RPRM*, *GADD45B*, *GADD45G*, and *SESN3*) and metastasis (*SERPINE1*, *ADGBR1*, and *THBS1*). The elevation of *NODAL*, *LEFTY1*, and *LEFTY2* in NT2/D1 cells and *LEFTY2* in TCam-2 indicates Nodal signalling is active 6 h post-activin A exposure. Notch signalling pathway components, including the receptor *NOTCH3* and the coactivator *MAML3*, were elevated by activin A in both NT2/D1 and TCam-2 cell lines, while readouts of Notch activity, *HEY1* and *HES1*, were only elevated in NT2/D1 cells. Activin A affected transcripts relating to activin signalling, increasing *SMAD3* and decreasing *INHBB* and *INHBA*. It also elevated those involved in BMP signalling inhibition, *SMAD6*, *SMAD7*, and *BAMBI*, in NT2/D1 cells, and lowered BMP4 target gene transcripts, *ID3* and *ID4*, in TCam-2 cells.

### 3.3. BMP4 Treatment

Only 96 transcripts (0.48%) were significantly altered in TCam-2 cells by 6 h of exposure to BMP4, compared with 1376 (6.9%) in NT2/D1 cells; 61 of these were common to both lines (Figure 2A). These shared BMP4 targets were associated with TGF-β, pluripotency, Hippo, HTLV-I infection, and Wnt pathways. Reflecting the synexpression of BMP signalling components [55], common transcripts also included BMP signalling inhibitors, *SMAD6, SMAD7*, and *BAMBI,* and known downstream targets inhibitors of DNA binding (*ID1/2/3/4*), *MSX2* [56], *SNAI2* [45], and *EVX1* [57]. BMP4 elevated *SMAD9* in both cell lines and *BMPR2* in NT2/D1 cells. Additional interesting transcripts modulated by BMP4 in both cells included *TGFΒ1*, *EGR1* (early growth Response 1), *KDM7A* (lysine demethylase 7A), and *HAND1* (heart and neural crest derivatives expressed 1). In BMP4-treated NT2/D1 cells, Wnt, MAPK, and PI3K-Akt signalling pathways were prominent amongst those affected. Downstream gene targets also encoded transcriptional effectors (NFKBIA, CREB5, and TIAM1) as well as proteins involved in cell survival (BDNF, FOS, and JUN) and cytoskeletal remodelling (RAC2) (Appendix A). Strikingly, BMP4 treatment elevated transcripts encoding several MAPK signalling pathway components only in NT2/D1 cells, including MAPKK kinases (MAP3K4 and MAP3K14) and phosphatases (DUSP2, DUSP5, DUSP6, and DUSP9). 

### 3.4. Common Targets of Activin A and BMP4

Both activin A and BMP4 elevated many transcripts in NT2/D1 cells encoding components linked with Hippo signalling, including transcription factors TEAD1 and FRMD6 (FERM domain-containing protein 6), the transcriptional repressor (DLG2), and target genes involved in anti-apoptotic or proliferative functions FGF1, SOX2, and AXIN2 (Figure 2A and Appendix A). 

Levels of 38 transcripts in TCam-2 cells and 523 in NT2/D1 cells were altered by both activin A and BMP4 exposures, indicating the potential for cross-talk between activin A and BMP4 pathways in testicular cancer cells (Figure 2B). Importantly, however, the transcriptional outcomes demonstrate that their activities would provoke distinct responses in each cell line. DAVID functional analyses showed that activin A and BMP4 modulate several common genes in TCam-2 cells that are primarily involved in differentiation, whereas in NT2/D1 cells, the potentially affected functions relate to transcript regulation, differentiation, neurogenesis, Wnt signalling, and angiogenesis (Figure 2B).

A fundamental feature of TGF-β superfamily signalling is the shared use of downstream signalling machinery. However, because there is an imperative to understand how germ cell tumours are affected by extrinsic factors, we also used Ingenuity Pathway software to explore the potential for upstream regulators to influence the outcomes of activin A and BMP4 signalling in these cells (Table 2 and Appendix A). We identified several key TGF-β superfamily components linked to common transcriptional outcomes, with *TGFB1* and *SMAD4* consistently identified. As stated earlier, signalling by multiple BMPs using SMAD4 would be expected to strongly influence BMP4 target expression, and this was robustly indicated in NT2/D1 cells. Environmental factors and competing signalling pathways were also identified. Common pharmaceuticals identified as potential upstream regulators of these signalling outcomes included aspirin, medroxyprogesterone acetate, estradiol, simvastatin, candesartan, curcumin, and tretinoin. In NT2/D1 cells, lipopolysaccharides, inflammatory signalling components, and cigarette smoke were notable potential upstream modulators. Several of these signalling molecules are of unequivocal importance for early germline development, including FGF2, KITLG, EGF, NRG1, and CXCL12.

### 3.5. Common and Distinct Regulation of Specific Genes by Activin A and BMP4

The validation of RNA-seq data using qRT-PCR is presented to illustrate the common and distinct responses of these two cell lines to activin A and BMP4 following 6 h of growth factor exposure (Figure 3A,B). *ID3* elevation by BMP4 and reduction by activin A was observed in both cell lines at 6 h. *IGFR1* and *MMP9* were significantly elevated in activin A-treated cells, whereas *KDM7A* levels were significantly higher following BMP4 treatment. *GREM2* was significantly elevated in activin A- treated TCam-2 cells, while in NT2/D1 cells it was higher following both activin A and BMP4 exposures. *PRDM14* was significantly decreased by both activin A and BMP4 in NT2/D1. Subsequent experiments over a longer time interval also used qRT-PCR analyses to document BMP4 target transcript levels following 48 h of growth factor exposure. *ID1* and *ID3* levels were robustly elevated in TCam-2 cells following 48 h of BMP4 treatment. Intriguingly, *SOX17* and *BAMBI*, implicated in yolk sac tumours [58], were increased in TCam-2 cells by both activin A and BMP4 exposure at 48 h.

### 3.6. Effects of Activin A on BMP Signalling Activation

To further investigate signalling cross-talk between activin A and BMP4 in TCam-2 and NT2/D1 cells, dual promoter luciferase assays were used to examine their signalling activities simultaneously [59]. Cells were infected with adenoviral reporter constructs, then stimulated with these factors. Activin A antagonized the BMP4-induced activation of BMP response element (BRE) in a dose-dependent manner in TCam-2 and in NT2/D1 cells, at both 6 h and 24 h (Figure 4A,B). In contrast, BMP4 did not impede the activin A-induced activation of the activin response element (CAGA) in TCam-2 cells but did result in the inhibition of NT2/D1 cells at the 24 hr timepoint at the higher dose (Figure 4C,D). Activin A exposure also impeded the BMP4-induced upregulation of BMP4 target genes *ID3* and *PRDM1* in TCam-2 cells in a dose-dependent manner, but it did not affect the levels of two other BMP4 target gene transcripts, *ID2* and *SOX17* (Figure 4E). 

### 3.7. Importins in TCam-2 and NT2/D1 Cells: A Role for IPO5 in BMP4 Responses

RNA sequencing analyses revealed that several importin-β family members were highly expressed in TCam-2 and NT2/D1 cell lines, most notably *IPO5*, previously shown to selectively transport BMP SMADs (SMAD 1/5/9) but not the activin SMADs (SMAD 2/3) [43] (Figure 5A).

To test whether IPO5 might enable BMP signalling in testicular cancer cells, siRNA was used to achieve transient *IPO5*-silencing in TCam-2 cells; silencing efficiency was determined by qPCR and Western blot. While *IPO5* mRNA levels were significantly reduced in both lines at 48 h and 72 h, the profile of IPO5 protein level reduction differed between the two. In TCam-2 cells, IPO5 was unaffected at 48 h but was significantly reduced at 72 h post-transfection; in contrast, IPO5 levels were lower in NT2/D1 cells at 48 h (*p* = 0.053) but not at 72 h (Figure 5B).

BMP4 target gene transcripts levels measured in *IPO5*-silenced and scram-transfected TCam-2 cells showed that *ID3*, *PRDM1*, and *SOX17* were significantly lower at 48 h post-BMP4 treatment in TCam-2 cells with siRNA treatment. NT2/D1 cell viability was severely reduced in the serum-free culture conditions required for this experiment, so the results of longer-term effects could not be obtained from this cell line under conditions used for the TCam-2 cells.

## 4. Discussion

Seminomas and embryonal carcinomas represent the vast majority of testicular germ cell tumours (TGCTs). This study reports the effects of activin A and BMP4 on the gene expression profiles of TCam-2 (representing seminoma cells) and NT2/D1 (representing non-seminoma cell type) cell lines and provides evidence that IPO5 may be important to selectively mediate BMP signals. RNA sequencing, bioinformatic analyses, and qRT-PCR were used to identify target genes and major signalling pathways altered by activin A and BMP4 in TCam-2 and NT2/D1 cell lines. Strikingly, activin A and BMP4 upregulated many growth factors, tyrosine kinase receptors (RTKs), and G protein-coupled receptors in both the cell lines, many of which are associated with the development of cancer. Notably, the insulin-like growth factor 1 receptor (*IGF1R)* was upregulated in both TCam-2 and NT2/D1 cells following activin A treatment. The IGFR is important for normal functioning of germ cells [60,61,62,63,64,65,66] and is highly upregulated in many TGCTs [67]. IGFR activation has been shown to increase the expression of the CXCR4 chemokine receptor, which is associated with cell survival and migration, including in the male germline [68,69,70] and matrix metalloproteins MMP2 and MMP9, associated with an invasive phenotype [71]. Interestingly, our data showed that transcripts encoding CCXR4 and MMP9 were upregulated in activin A-treated NT2/D1, whereas *MMP2* and *MMP9* were upregulated in activin A-treated TCam-2 cells, suggesting the potential for activin A to induce more active IGF1R signalling in both cell lines.

Activin A treatment upregulated several key modulators of pathways that signal through RTKs, such as Ras/Raf/MAPK and PI3 kinase. The upregulation of RAS [72], PI3 kinase [73], and MAP kinase [74,75] pathways has been demonstrated in both seminomas and non-seminomas. Interestingly, AKT3, a key PI3K pathway modulator shown here to be upregulated by activin A in TCam-2 cells, was identified to be frequently overexpressed in TGCTs and associated with poor survival outcomes [34]. Our data also indicate that activin A and BMP4 treatment modulates the expression of transcription factors, effectors, and targets genes involved in the Ras/Raf/MAPK and PI3K pathways. While activin A modulated several genes involved in MAPK signalling pathway in both cell lines, the impact of BMP4 treatment upregulating transcripts for several MAPK signalling pathway components was primarily documented in NT2/D1 cells. These influence processes affecting cell migration, including cytoskeletal remodelling, endocytosis, and cell mobility, in alignment with our observation that these factors increased TCam-2 cell migration [45]. p53 signalling pathway components, involved in mediating responses to DNA damage induced by agents such as cisplatin, were also altered in both cell lines by activin A. The activation of the p53 signalling pathway in TGCTs was previously shown to lead to apoptosis rather than cell cycle arrest [76]. This accords with the upregulation of the transcript encoding the executioner caspase, *CASP3,* in activin A-treated NT2/D1 cells. In both cell lines, activin A also upregulated the core components of the Notch signalling pathway, which, when constitutively active, results in migration defects and premature differentiation, and results in germ cell loss when active in Sertoli cells [77,78].

Amongst the novel BMP4 targets identified were the transcript encoding the dual histone demethylase, KDM7A, previously shown to promote tumour growth and mediate androgen receptor activity, and FGF4 [79,80]; both were upregulated in both cell lines following BMP4 stimulation. FGF4 supports germ cell survival/proliferation [81,82], and its expression in testicular cancer has been reported [83,84,85]. Activin A and BMP4 treatments elevated transcripts encoding several important immune-related molecules in TCam-2 and NT2/D1 cells, notably the proinflammatory cytokine IL6 and its receptor, IL6R. Elevated IL6 levels are documented in seminomas [86,87], while dysregulated IL6 expression features in many cancers and is associated with tumour growth and invasiveness [88]. In this context, it is interesting to note that compounds with anti-inflammatory properties, such as aspirin and curcumin, were predicted to have inhibitory effects on activin A signalling in both cell lines. Aspirin has been shown to inhibit cell proliferation and its effects can be mediated by TGFB1 [89,90]. The *TGFB1* transcript was upregulated in both cell lines following BMP4 exposure and in NT2/D1 cells following activin A exposure. TGF-β1 was also predicted as a common upstream regulator of both activin A and BMP4 signalling. The higher expression of *TGFB1* has been reported in several cancers [91], including testis cancer [87]. TGF-β1 acts both as a tumour suppressor in initial cancer stages and as an oncogene in later stages, driving epithelial to mesenchymal transition (EMT) and tumour metastasis [92]. In NT2/D1 cells, both activin A and BMP4 treatments significantly altered Hippo signalling pathway, an important pathway that controls cell growth and differentiation, often dysregulated in a number human cancers [93]. The key transcriptional factor and readout of the Hippo signalling pathway, TEAD1, was significantly upregulated following activin A and BMP4 stimulation, where as its coactivator TAZ/WWTR1 was upregulated following activin A treatment in NT2/D1 cells. Both TEAD and TAZ play a central role in Hippo signalling-mediated tumorigenesis and the overexpression or activation of factors are linked to tumour initiation and metastasis [94,95]. These transcriptional changes, following short term exposures in testicular cancer cell lines, demonstrate how both activin A and BMP4 can influence multiple signalling pathways that modify cell migration, survival, and cell cycle progression behaviours in TGCTs.

The identification of several transcripts and signalling pathways altered by both activin A and BMP4 treatments highlights points where both synergistic and antagonistic outcomes could emerge from cells exposed to inputs from the two arms of TGF-β signalling. These investigations using a dual luciferase reporter assay demonstrated that activin A dose-dependently inhibited BMP4-mediated BMP response element (BRE) activation in TCam-2 and NT2/D1 cell lines. In contrast, BMP4 had limited effects on activin A-induced promoter (CAGA) activation, although there was some degree of inhibition in NT2/D1 cells. Validating qPCR analysis showed that activin A impeded the upregulation of established BMP4 target genes *ID3* and *PRDM1* following BMP4 stimulation. This may be of particular relevance in the context of evidence that BMP signalling inhibition and decreased levels of its targets (e.g., SOX17) are key to the early steps in reprogramming TCam-2 cells into an EC-like cell type [11]. However, not all BMP4 target genes tested in TCam-2 cells in this study, including *SOX17*, were affected by activin A treatment. Here, it is important to note that, unlike the aforementioned studies carried out over several weeks, our goal was to identify the early targets of activin A and BMP4 in testicular cancer cells to investigate the initial stages of pathway crosstalk, and thus the treatments were short term (≤48 h). It will be of interest to ascertain the time course over which activin A exerts inhibitory effects on BMP signalling; we predict this could reveal stages of tumour development that are sensitive to manipulation or vulnerable to environmental exposures relating to changes in TGF-β pathway signalling.

Our data showed that Activin A altered many more transcripts in TCam-2 cells than in NT2/D1 cells, while BMP4 altered expression of many more genes in NT2/D1 cells relative to TCam-2 cells. This could be due to intrinsic differences in genomic or proteomic landscape between the two cell lines [96,97]. In this study, we further explored the importance of the high levels of *IPO5* transcripts in NT2/D1 cells, as IPO5 selectively transports BMP-specific SMADs, SMADs 1/5/9, but not those generally involved in activin/TGF-β/nodal signalling, SMADs 2/3 [43]. In recent years, IPO5 has attracted much attention in the fields of cancer and virology. *IPO5* is elevated in colorectal cancers and oesophageal cancer, and this has been associated with increased tumorigenicity and metastasis [98,99]. Elevated IPO5 levels in the serum of oesophageal and cervical cancer patients has been proposed as a cancer biomarker [100]. We had previously demonstrated that IPO5 transcript and protein are abundant in foetal germ cells [40,42], GCNIS, seminomas, and in TCam-2 cells [45]. Further, the siRNA reduction of *IPO5* in TCam-2 cells lowered levels of *SNAI2*, a BMP4-regulated gene expressed in TGCTs known to function in cancer metastasis [45]. The present results demonstrate that *IPO5* knockdown in TCam-2 cells impeded the expression of additional BMP targets, such as *ID3, SOX17*, and *PRDM1* (PR domain containing 1, previously called BLIMP1). These findings illustrate that IPO5 can selectively modulate BMP signalling in TCam-2 cells and thus may determine the fate of testicular germ cell tumours. Although NT2/D1 cells exhibit high levels of *IPO5,* the experimental conditions used in our study did not support the analysis of IPO5 in mediating BMP signalling in these cells. Given the differences in how each cell responds to activin A and BMP4 stimulation, we propose that the persistent or induced elevation of IPO5 may be a feature enabling the emergence of pluripotent embryonal carcinoma cells from testicular germ cells.

In summary, this study identified signalling pathways and novel signalling components that were altered in testicular cancer cell lines stimulated with activin A and BMP4, demonstrating the potential for antagonism and synergy between the two arms of TGF-β signalling in testicular cancer cells. Furthermore, our study reinforced the involvement of IPO5 in mediating BMP signalling in seminoma cells. These outcomes (summarized in Figure 6) support the accumulating evidence that local changes in TGF-β signalling components, including ligands, receptors, and inhibitors, will have implications for testicular germ cell tumour aetiology and progression.

## Figures and Tables

**Figure 1 cells-12-01000-f001:**
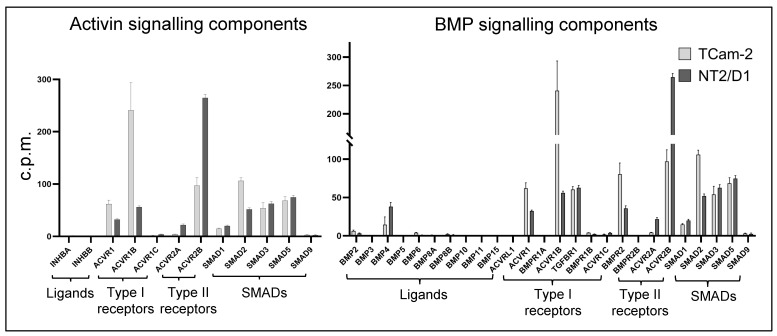
**TGF-β superfamily signalling pathway transcript levels.** Transcript levels encoding activin and BMP signaling components identified in [54] differ between the two testicular cancer cell lines. Data are the mean of counts per million (cpm) in control cells (cultured in vehicle only) from four independent RNA-seq experiments (n = 4); error bars indicate standard error of the mean (SEM).

**Figure 2 cells-12-01000-f002:**
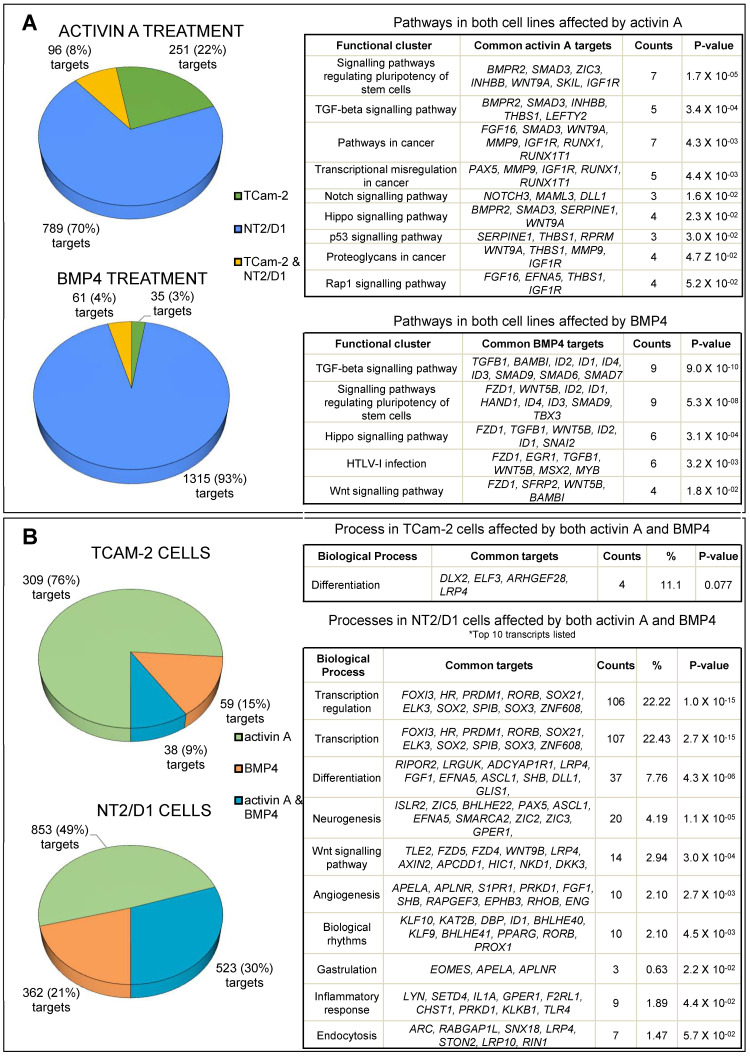
**RNA-seq analyses of activin A- and BMP4-treated TCam-2 and NT2/D1 cells.** (**A**): Transcript levels changes in TCam-2 cells (347 significantly different) and NT2/D1 cells (885) resulting from 6 h of 5 ng/mL activin A exposure identified in RNA-seq data, with 96 transcripts altered in both cell lines. Significant transcript level changes in TCam-2 cells (96) and NT2/D1 cells (1376) were identified after 6 h of 5 ng/mL BMP4 exposure, with 35 transcripts altered in both. (**B**): Common transcripts affected by both activin A and BMP4 in TCam-2 cells (38) and NT2/D1 cells (523). The KEGG pathway analysis of these common differentially expressed genes used the DAVID online tool, with FDR cut-off set at 0.1. Appendix A contain the full DAVID analysis of the differentially expressed genes and the signalling pathways affected, respectively.

**Figure 3 cells-12-01000-f003:**
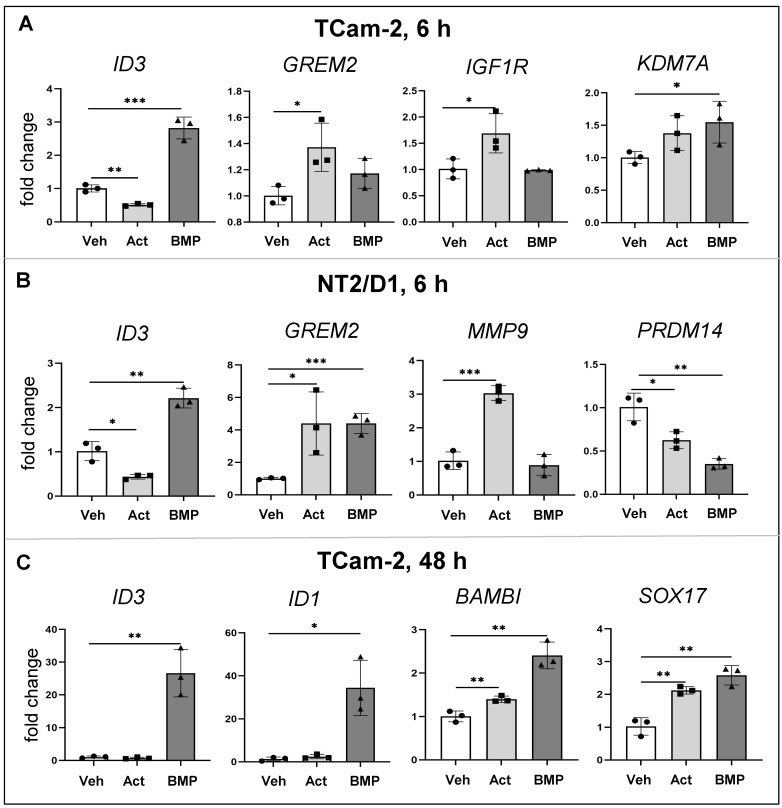
**Validation of selected transcripts in TCam-2 cells and NT2 cells.** Independent cultures of TCam-2 (**A**,**C**) and NT2/D1 (**B**) cells were stimulated with vehicle, 5 ng/mL activin A or 5 ng/mL BMP4 (indicated on the Y-axis) in serum-free medium for the indicated time and transcript levels measured by qRT-PCR. Ct values were normalized to *RPLP0* and treatment outcomes expressed as fold-change relative to the vehicle control, which was set to one. Individual data points show outcomes from three independent experiments (n = 3); error bars indicate SEM. Statistical significance determined by unpaired *t* test; *** indicates *p* ≥ 0.001; ** indicates *p* ≥ 0.01; * indicates *p* ≥ 0.05.

**Figure 4 cells-12-01000-f004:**
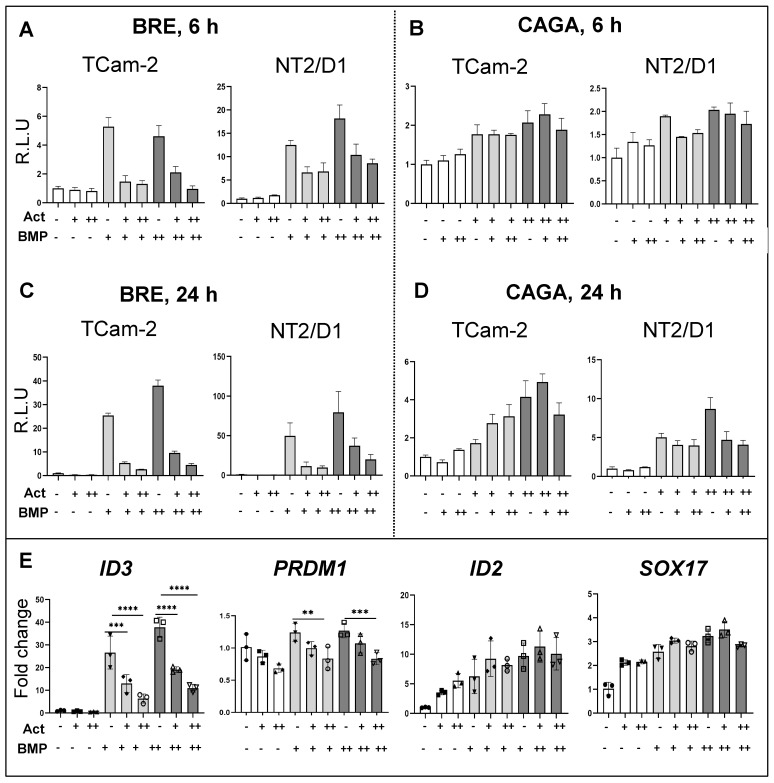
**Transcript levels and signalling outcomes show activin A affects BMP signalling activation.** TCam-2 and NT2/D1 cells infected with the adenoviral BMP response element reporter construct, Ad-Bre-F-Luc (BRE), and adenoviral activin response element reporter construct, Ad-CAGA-G-Luc (CAGA), were stimulated with activin A (Act) or BMP4 (BMP) at 5 ng/mL (low dose, denoted by ‘+’) and 25 ng/mL (high dose, ‘++’) or by both factors at the indicated doses. Luciferase activity was measured at 6 and 24 h after factor addition. All treatments were carried out in triplicate. Vehicle control readings were normalized to 1; all other readings were normalized to vehicle controls. (**A**–**D**): **representative outcomes** from one experiment following BRE and CAGA luciferase measurements at 6 and 24 h after growth factor addition are presented as mean ± SD. (**E**): After the stimulation of TCam-2 cells with activin A or BMP4 at 5 ng/mL (‘+’) and 25 ng/mL (‘++’) or a combination at indicated dose for 48 h, samples were collected to measure BMP4 target gene transcripts. Ct values normalized to *RPLP0* are expressed as fold-change relative to vehicle control. Individual data points show outcomes from three independent experiments (n = 3); error bars indicate SEM. Statistical significance determined by ordinary one-way ANOVA; **** *p* ≥ 0.0001; *** *p* ≥ 0.001; ** *p* ≥ 0.01.

**Figure 5 cells-12-01000-f005:**
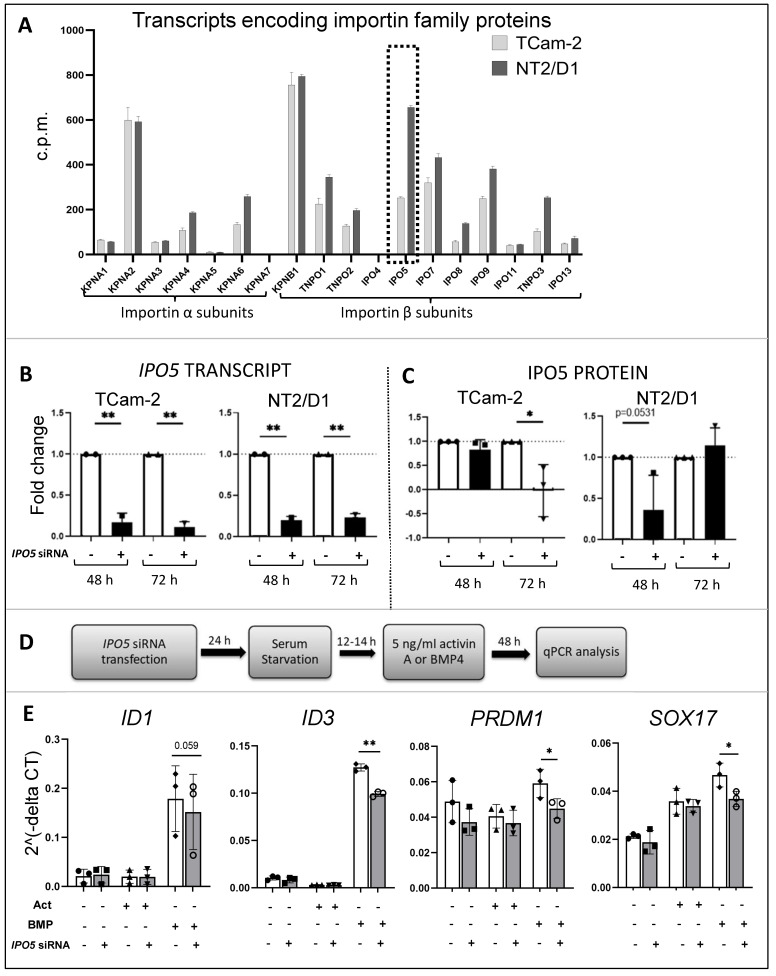
**Importins in TCam-2 and NT2/D1 cells, and a role for IPO5 in BMP4 responses.** (**A**): Transcripts encoding importin proteins were present in TCam-2 and NT2/D1 cells at relatively equivalent levels; however, *IPO5* (dotted box) was >2-fold higher in NT2/D1 compared to TCam-2 cells. Data are the mean cpm in vehicle-treated (control) cells from four independent experiments (n = 4); error bars indicate SEM. (**B**,**C**): The qRT-PCR and Western blotting measurements of *IPO5* mRNA and protein levels at 48- and 72 h post-transfection with *IPO5* siRNA or scram siRNA. Data are from three independent experiments (n = 3). Protein band values normalized to loading control and *IPO5* siRNA outcomes are shown as fold-change relative to the scram siRNA value set to 1. (**D**): Protocol used to assess contribution of IPO5 to BMP4 signalling in TCam-2 cells. (**E**): *IPO5* knockdown reduced the BMP4-induced upregulation of BMP target genes *ID1*, *ID3, PRDM1*, and *SOX17* in TCam-2 cells. All values are normalised to *RPLP0*. 2^ (-delta CT) values were compared between scram siRNA and IPO5 siRNA transfected cells following activin A and BMP4 treatment. Data are mean of three independent experiments (n = 3); error bars indicate SEM. Significance was determined by a paired two-tailed *t* test; ** *p* ≥ 0.01; * *p* ≥ 0.05.

**Figure 6 cells-12-01000-f006:**
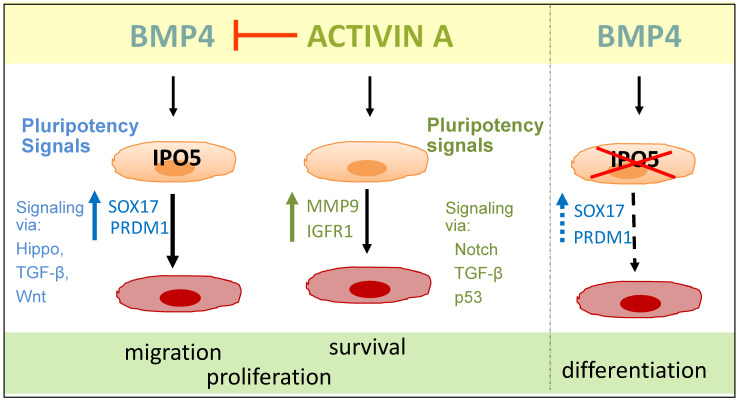
**Summary:** Both Activin A and BMP4 modulate a number of different signalling pathways that can alter migration, cell proliferation, survival, and differentiation in germ cells; ultimately determining the tumorigenicity of the cells. The elevation of activin A can dampen the BMP signalling response, presumably by competing for access to receptors and intracellular signalling molecules. Decreasing nuclear protein, IPO5 levels can reduce BMP4 responsiveness. Thus, based on the existing data, we propose that BMP4 signalling, which is integral to maintenance of the seminoma cell characteristics of TCam-2 cells, can be disrupted by a local increase in activin A bioactivity or by a stimulus, as yet undefined, which reduces the levels of IPO5 in these cells. Alternatively, the elevation of IPO5 expression, as seen in a number of different cancers, may be a contributing factor to tumorigenesis by increasing the BMP signalling activity and BMP responsiveness in these cells.

**Table 1 cells-12-01000-t001:** Primer sequences used for qRT-PCR.

Gene	Accession No.	Forward Primer (5′ to 3′)	Reverse Primer (5′ to 3′)
*IPO5*	NM_178310	AGGTCCTTCCACACTGGTTG	AATTGCCTCGTGCATTTCTC
*GREM2*	NM_022469.4	GCTGATGTGTTCCTGACCGA	TGATCCACCGCCTGGTTTAG
*NOTCH3*	NM_000435.3	ATGGTATCTGCACCAACCTGG	GATGTCCTGATCGCAGGAAGG
*MMP9*	NM_004994.3	CAGTCCACCCTTGTGCTCTT	CGACTCTCCACGCATCTGTG
*ID1*	NM_002165.4	CGAGGCGGCATGCGTT	ACGTAATTCCTCTTGCCCCC
*ID2*	NM_002166.5	CCGTGAGGTCCGTTAGGAAA	AGCTTGGAGTAGCAGTCGTT
*ID3*	NM_002167.5	AGCGCGTCATCGACTACATT	TGACAAGTTCCGGAGTGAGC
*ID4*	NM_001546.4	GTGCGATATGAACGACTGCT	TGCTGACTTTCTTGTTGGGC
*PRDM1*	NM_001198.4	TACATACCAAAGGGCACACG	TGAAGCTCCCCTCTGGAATA
*PRDM14*	NM_024504.4	CAGAGGGAGCCTCTCTACGAT	GGACGTGGGGAATTGGGTA
*SOX17*	NM_022454.4	GGACCGCACGGAATTTGAAC	GGACACCACCGAGGAAATGG
*RPLP0*	NM_001002.3	CTATCATCAACGGGTACAAACGAG	CAGATGGATCAGCCAAGAAGG

**Table 2 cells-12-01000-t002:** Potential upstream regulators of outcomes from activin A and BMP4 signalling identified using Ingenuity Pathway Analysis. The Z score cut off of ±1.8 was employed, with a Z-score ≤ -1.8 indicating inhibition and a Z-score ≥1.8 indicating activation. The predicted upstream regulators were categorized as: TGF-ß signalling components, other signalling components, or pharmaceuticals. (A): Key upstream regulators predicted to affect either activin A or BMP4 targets in each cell line. (B): Key upstream regulators predicted to affect both activin A and BMP4 signalling targets in either the TCam-2 or NT2/D1 cell line. The bold font denotes Z score differences >1 between the groups. Appendix A contains the full list of identified upstream regulators.

A						B					
Upstream Regulators Affecting Activin A Targets in Both Cell Lines			Upstream Regulators Affecting BMP4 Targets in Both Cell Lines			Upstream Regulators Affecting Activin A and BPM4 Targets in NT2 Cells			Upstream Regulators Affecting BMP4 and Activin A Targets in TCam-2 Cells		
	Z-Score in NT2/D1	Z-Score in TCam-2		Z-Score in NT2/D1	Z-Score in TCam-2		Z-Score Activin A	Z-Score in BMP4		Z-Score in Activin A	Z-Score in BMP4
TGF BETA SIGNALLING COMPONENTS						TGF BETA SIGNALLING COMPONENTS					
SB-431542	−2.521	−2.207	BMP7	2.214	2.359	**BMP4**	**1.941**	**5.063**	**TGFB1**	**3.392**	**1.984**
SMAD2	1.953	1.98	BMP15	2.443	2	**BMP2**	**2.192**	**3.885**	SMAD4	1.844	1.972
SMAD4	2.347	1.844	BMP10	2.583	1.976	**BMP6**	**2.216**	**3.254**			
TGFB3	2.398	2.565	**BMP6**	**3.254**	**1.974**	**SMAD4**	**2.347**	**3.89**			
SMAD3	2.578	2.422	**BMP2**	**3.885**	**2.478**	TGFB3	2.398	2.214			
ACVR1C	2.63	1.982	**SMAD4**	**3.89**	**1.972**	SMAD3	2.578	2.325			
Tgf beta	3.247	2.378	**TGFB1**	**3.934**	**1.984**	Tgf beta	3.247	2.624			
**TGFB1**	**4.435**	**3.392**	**BMP4**	**5.063**	**2.767**	TGFB1	4.435	3.934			
**OTHER SIGNALLING MOLECULES**						**OTHER SIGNALLING MOLECULES**					
NR3C2	1.953	1.953	IGF1	1.911	2.155	LIF	1.811	1.803	EGF	2.5	2.563
IL1B	2.725	2.064	Insulin	2.019	2.158	NRG1	2.035	2.498			
IL1A	2.748	1.925	IL6	2.092	2.578	CXCL12	2.104	2.468			
**NFKBIA**	**3.015**	**1.8**	CXCL12	2.468	1.934	OSM	2.165	1.952			
**EGF**	**3.586**	**2.5**	GDF9	2.574	2.236	EGR1	2.587	1.818			
**TNF**	**4.395**	**3.184**	IFNG	2.671	1.988	KITLG	2.595	2.599			
			FGF2	2.724	2.378	IL1B	2.725	2.122			
			ESR2	2.789	1.912	IL1A	2.748	1.843			
			**PDGF BB**	**3.274**	**1.943**	EGF	3.586	4.007			
			**GDF2**	**3.63**	**2.376**	RELA	3.994	3.257			
			**EGF**	**4.007**	**2.563**	TNF	4.395	4.034			
**PHARMACEUTICALS**						**PHARMACEUTICALS**					
aspirin	−3.323	−1.953	beta-estradiol	2.697	2.632	aspirin	−3.323	−3.035	beta-estradiol	2.15	2.632
curcumin	−1.99	−1.951	**MPA**	**3.097**	**2.449**	triptolide	−2.538	−2.111	MPA	1.947	2.449
tretinoin	2.796	3.35				simvastatin	−2.412	−1.937			
**MPA**	**3.446**	**1.947**				**candesartan**	**−1.948**	**−1.948**			
beta-estradiol	1.957	2.15				TSH	1.901	1.901			
						beta-estradiol	1.957	2.697			
						ciprofibrate	1.98	1.98			
						GnRH-A	1.987	2.216			
						deferoxamin	2.137	2.9			
						**tretinoin**	**2.796**	**4.597**			
						MPA	3.446	3.097			

## Data Availability

Contact corresponding authors.

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
