# Peer review of "Activin and BMP Signalling in Human Testicular Cancer Cell Lines, and a Role for the Nucleocytoplasmic Transport Protein Importin-5 in Their Crosstalk"

_cells, 2023, doi:10.3390/cells12071000_

Round 1

Reviewer 1 Report

The article entitled ‘Activin and BMP signaling in human testicular cancer cell lines, and a role for the nucleocytoplasmic transport protein Importin-5 in their crosstalk’ describes the Identification of  signalling pathways such as Hippo, Notch etc by Activin or BMP signalling responses. The article is very intriguing, well written, and provides compelling evidence.  However, the author could improve the article.

1.      The Figure 1A, needs better clarity of the image, some of the colours are saturated in the background.

2.      Figure 1 B is too small, the font size of the text needs to be increased in the figure.

3.      The gene expression data depicted in Figure 4, can be confirmed at the protein level with immunoblots.

4.      Figure 4, the authors need to describe if the fold change of the genes is normalized to the house keeping gene or not. Gene of interest/housekeeping gene should be included on the y axis and in the figure legend.

5.      Why NT2/D1 cells were not tested at 48 hrs? It would be interesting to know expression of the genes at later time points.

6.      Number of experiments needs to be indicated in the figure legend on Figure 4, Figure 5.

7.      Statistical analysis (ANOVA) needs to be performed in Figure 5 A, B, C, D. The statistical significance between the error bars is missing (p value).  

8.      The authors need to confirm the role of importins using other methods like nucleo-cytoplasmic fractionation or Immunofluorescence in the presence or absence of Importins inhibitors to know if they paly a role nucleocytoplasmic shuttling and how long does it take for this phenomenon to take place upon activation of Activin or BMP signalling to support the claims of figure 6.

9.      Fig 6B, where IPO5 siRNA treatment was performed need to be complemented with  data where Wild type IPO5 plasmid is transfected to understand if the genes regulated can be rescued.

10.  The authors mentioned Delta Delta CT method for Fig 6D, can the authors clarify why it was not used for calculating gene expression of other genes in Fig4, and 5.

11.  Can the authors probe for biological roles such as invasion, migration of the pathway as it is unclear?

12.  Can the authors include a summary picture to give an overview of the findings.

13.  The discussion needs to include other related articles.

Sanchez-Duffhues G, Williams E, Goumans MJ, Heldin CH, Ten Dijke P.             Bone morphogenetic protein receptors: Structure, function and targeting by selective small molecule kinase inhibitors.

Mu Y.  Non-Smad signaling pathways.

Reviewer 2 Report

Radhakrishnan et al. focused on the roles of activin and BMO signaling in human testicular cancer cell lines. In general, I think the authors did quite some work, and the results are convincing and interesting. The analysis pipeline that the authors used is sophisticated and impeccable. Although some details need further explanation, I have some questions related to the methods and results. Here are some questions and comments that the authors should think about as they make changes to their work.

1.     Figure 1 is a cellular substructure map of the factors related to the TGFB pathway. According to the introduction, this figure is the result of a literature review. It does not need to appear in the text but can be placed in the supported figure. In addition, the relevant factors that the manuscript focuses on are not highlighted in this figure. The deletion of this figure does not affect the integrity of the introduction section.

2.     The introduction part does not well clarify the background and the necessity of the research. TGFB is strongly associated with the occurrence of tumors, and the related changes of TGFB can be observed in any tumor. The third paragraph begins with the relationship between gonadal cancer and the TGFB pathway, and the fourth paragraph describes the related research on the BMP signaling pathway and activin A in cell lines. This indicates that there isn't a compelling argument in favor of TGFB research in gonad cancers. 

3.     The TCam-2 cell line used in this paper is not a universal cell line, and the source of this cell needs to be confirmed. It can be determined by genome characteristics or a special diagnosis marker. Otherwise, it is impossible to confirm whether the results of this study are related to testicular cancer.

4.     What does "Media was changed" in line 160 mean? I guess "media" may be " medium ", and "changed" may represent refresh. The use of relevant terms should be standardized and rigorous.

5.     With regard to cell culture, why should we remove serum in the later stage of the growth factor exposure experiment? Even though the experiment with the blank control is also done after the serum is removed, the serum removal may cause a stress reaction in the cells, and the data from the experiment may show that the cells are in an abnormal physiological state. In what way does the author view this? 

6.     The paragraph font in Section 2.2 of Materials and Methods is obviously strange. 

7.     Because it is a table, I advise replacing "Figure 3" with "Tables". 

8.     As for Figure 4, the expression of the same transcript should be compared horizontally in the treatment of different cell lines at different times; otherwise, the correlation between the treatment time and the expression level cannot be shown in the figure.

9.     With regard to the luciferase experiment, the MOI of the two adenovirus vectors is not the same. This may cause exceptions in the Luciferase report results. It can be seen that the ordinate units of the A-D sub-graph in figure 5 are not consistent, and there is a difference of nearly 10 times. It is unclear why different MOIs should be used. Ask the author to explain.

10.  In enrichment analysis, it was observed that the Hippo signaling pathway, the TGFb pathway, and other pathways were enriched. However, there is no corresponding discussion in the discussion section. Under the density-dependent selection experiment of HeLa cells, Tao Li et al. observed a Hippo signaling pathway and TGFb pathway enrichment difference in the differentially expressed genes between two different environment-adapted cell lines, which was related to cell survival (PMC8288455). The important function of the Hippo pathway in normal tissues is organ-size control. Tumor cells may lose or have abnormal regulation of this pathway, resulting in tumor proliferation. In the discussion, the author also mentioned that cell migration and survival are related to activin A and BMP4. The size of organs is also closely related to cell density, and cell migration can also be seen as the process of cells moving from high-density areas to low-density areas. Both this project and the article by Tao Li et al. have observed changes in the Hippo pathway, indicating that the Hippo pathway may also participate in cell migration. These should be included in the discussion section.

Round 2

Reviewer 2 Report

The authors responded to all of my concerns carefully. I have no further questions.